# Biological Control May Fail on Pests Applied with High Doses of Insecticides: Effects of Sub-Lethal Concentrations of a Pyrethroid on the Host-Searching Behavior of the Aphid Parasitoid *Aphidius colemani* (Hymenoptera, Braconidae) on Aphid Pests

Armando Alfaro-Tapia [1,2], Jeniffer K. Alvarez-Baca [1,2], Eduardo Fuentes-Contreras [2,3] and Christian C. Figueroa [1,2,*]

1   Instituto de Ciencias Biológicas, Universidad de Talca, 1 Poniente 1141, Talca 3460000, Chile; aalfaro@utalca.cl (A.A.-T.); jealvarez@utalca.cl (J.K.A.-B.)
2   Centre for Molecular and Functional Ecology in Agroecosystems, Universidad de Talca, Talca 3460000, Chile; efuentes@utalca.cl
3   Facultad de Ciencias Agrarias, Universidad de Talca, Talca 3460000, Chile
*   Correspondence: alfigueroa@utalca.cl

**Abstract:** The use of synthetic insecticides may cause failures in the biological control of insect pests due to undesired side effects on natural enemies and the rapid evolution of insecticide resistance in agroecosystems. Residues of neurotoxic insecticides can interfere with the recognition of chemical cues used by natural enemies to find pests. We investigated the effects of sub-lethal concentrations of the pyrethroid lambda-cyhalothrin on the interaction between the aphid parasitoid wasp *Aphidius colemani* and the peach potato aphid *Myzus persicae*. We studied changes in host-searching and oviposition behavior through laboratory bioassays when susceptible and *kdr*-resistant aphids are offered to parasitoid females, evaluating the effect of applying insecticides on the interacting species. The patch residence time, exploration, oviposition, and grooming were significantly disturbed when the parasitoids were offered resistant aphids sprayed with sub-lethal doses, but not when the parasitoids were offered susceptible *M. persicae* exposed to sub-lethal doses. We discuss how the effects of insecticides on parasitism behavior may result in failures of biological control if natural enemy populations are not adequately managed, particularly for the management of insecticide-resistant pest populations. Efforts to introduce biological control in integrated pest management (IPM) programs are also discussed.

**Keywords:** parasitism behavior; aphid–parasitoid interaction; insecticides; pyrethroids; sub-lethal effects

## 1. Introduction

Biological control is a significant component of integrated pest management (IPM) programs, but synthetic insecticides are still commonly used to control many pest populations [1]. This is the case with the peach potato aphid, *Myzus persicae* (Sulzer), a cosmopolitan and highly polyphagous aphid responsible for significant economic losses in several crops [2]. The use of insecticides in *M. persicae* has favored a complex and rapid evolution of a broad array of resistance mechanisms to almost all kinds of insecticides through enhanced detoxifying enzymes, insensitivity mutations [3,4], cuticular thickening, and sequestration [5]. The most common mechanism of target-site mutation is known as 'knockdown resistance' (kdr), which confers resistance to pyrethroids and DDT caused by a single amino acid substitution (L1014F) in the voltage-gated sodium channel [6]. However, insecticide resistance involves fitness costs in insecticide-free environments. Individuals carrying resistant alleles show significantly lower overwintering survival, reduced

responses to aphid alarm pheromones, and increased vulnerability to parasitoids than homozygous susceptible individuals [7,8]. For instance, *M. persicae* aphids carrying the *kdr* mutation have a reduced response to the alarm pheromone (E)-$\beta$-farnesene, making aphids more prone to being parasitized [7,9].

This situation urgently calls for the adoption of methods that limit the development of resistance in aphid populations and reduce the side effects of insecticides on beneficial arthropods [10], particularly on their natural enemies (e.g., aphid parasitoid wasps), which otherwise will continue at a low frequency/diversity in agroecosystems [11]. Classical conservation and augmentative biological control programs have been implemented to control *M. persicae* populations, primarily based on Braconid parasitoid wasps, such as *Aphidius colemani* Viereck [12,13]. The outcome of this aphid–parasitoid interaction and the success of the biological control of the peach potato aphid has been reported as largely dependent on the life history strategies of both interacting insects [14].

It is well known that insecticides modify agronomically important traits in aphid parasitoids and their target aphids. On the one hand, sub-lethal doses of insecticides produce loss of reproductive performance, disorientation, and unusual foraging behaviors in aphid parasitoids [15–18]. The neurotoxicity of insecticides reduces the recognition of chemical cues emitted by the plant–aphid interaction, altering parasitism behavior [19]. Indeed, some studies have shown that sub-lethal doses of insecticides have detrimental effects on host-finding and oviposition performance in parasitoids [19–23]. However, the effects can disappear after 24 h of exposure [15]. In contrast, other studies have shown that some variables of host finding behavior (e.g., patch time allocation) are not significantly disturbed by sub-lethal doses of insecticides, attributed to some level of insecticide tolerance developed by parasitoid wasps [16,24].

As many insecticides have lethal or sub-lethal effects on aphid parasitoids [21,25], more knowledge on the impact of insecticides on parasitoid behavior is needed for integrating biological control with chemical control [21] so that components of parasitism behavior can be used to predict the success of biological control [26]. The balance between sub-lethal effects of insecticides on aphid parasitoids and the fitness cost associated with insecticide resistance in aphids may define antagonistic or complementary results from both managing strategies [23]. Hence, finding a balance between the use of insecticides to control aphids and their side effects on aphid parasitoids appears to be a key task in IPM programs [27]. The idea of studying sets of parasitoid behaviors and correlating them across situations (e.g., with or without insecticides, parasitizing insecticide-resistant or susceptible aphids) represents a major shift in addressing the outcome of aphid–parasitoid interaction. In essence, if parasitoid and aphid traits show a high correlation, they should be studied as a package rather than as isolated units [28].

In this paper, we addressed the effects of sub-lethal doses of lambda-cyhalothrin (a pyrethroid insecticide) on the patch time allocation and host-finding behaviors in the aphid parasitoid wasp *A. colemani* when mated females were faced with susceptible and *kdr*-resistant individuals of the peach potato aphid under laboratory conditions.

## 2. Materials and Methods

### 2.1. Aphids

Individuals of the peach potato aphid *M. persicae* were collected from potato (*Solanum tuberosum* L.) fields along a north–south transect ranging from Region Metropolitana 33° S to Osorno 40° S. Aphids were repeatedly sampled alive from the same parthenogenetic colonies from September 2014 to January 2015. Aphid clonal lineages were separately obtained by multiplying one single initial parthenogenetic female on potted sweet pepper *Capsicum annuum* var. *grossum* (cv. Resistant). The aphid lineages were reared enclosed in a transparent plastic cylinder covered with mesh cloth fabric on top to facilitate air circulation and prevent the aphids from escaping. Each aphid clonal lineage was genotyped at eight microsatellite loci (*Myz2*, *Myz3*, *Myz9*, *Myz25*, *M35*, *M37*, *M40*, and *M63*) previously described [29,30]. The identification of insecticide resistance mutations was performed

using the TaqMan assay in a STRATAGENE MX 3000 (Agilent Technologies, Santa Clara, CA, USA) thermocycler, screening for those mutations present in *M. persicae* populations in Chile [30]. The *kdr*, super-*kdr,* and MACE mutations were identified according to Anstead et al. [31,32]. Additionally, the constitutive activity of carboxylesterases (EST) was determined in all clonal lineages following the microplate bioassay described by Devonshire [33]. Finally, the presence of different endosymbionts known to occur in aphid populations [34] and previously reported conferring protection against parasitoids in aphids [35] was tested using the amplification by Polymerase Chain Reaction (PCR) of 16S rDNA from whole-body aphid DNA as described in Peccoud et al. [36]. This method allows screening for the endosymbiont bacteria *Hamiltonella defensa*, *Regiella insecticola*, *Serratia symbiotica*, *Rickettsia*, *Rickettsiella,* and *Spiroplasma* [34]. Therefore, two *M. persicae* genotypes, one susceptible to insecticides and one homozygous for *kdr* with the same endosymbiont (*R. insecticola*), were chosen for subsequent bioassays.

### 2.2. Parasitoids

*Aphidius colemani* wasps were obtained by sampling mummies and alive *M. persicae* aphids on conventional peach orchards in Central Chile (Maule Region) during February 2015. In Chile, conventional management includes constant monitoring and, in the presence of the first aphids' outbreak, applications of pyrethroids and neonicotinoids should be made [37]. Live sampled aphids were determined as *M. persicae* under a binocular microscope following taxonomic keys [38] and reared in the laboratory on sweet pepper plants until parasitized individuals formed mummies. The emerged parasitoids were determined using taxonomic keys [13], and they were used for founding laboratory colonies of *A. colemani*. Parasitoids were reared on *M. persicae* aphids established on sweet pepper plants grown in pots (10 cm diameter Ø) in acrylic ventilated cages (55 × 50 × 55 cm). Seedlings (eight to ten young leaves) of sweet pepper were grown in seed trays on Turba Sunshine peat moss as substrate and irrigation following routine practices; pesticide applications were strictly prohibited. Sweet pepper plants infested with 2nd and 3rd instar aphid nymphs were produced five days after the release of 40–50 *M. persicae* parthenogenetic aphids inside the acrylic cages; then, infested plants were transferred into parasitoid rearing cages. Parasitoids were fed with a honey solution at 30% and water *ad libitum*. For the experiments, individual mummies sampled from laboratory colonies were isolated in Petri dishes containing small sections of a leaf (2.0 cm Ø). They were checked for adult parasitoid emergence twice a day. All individuals that emerged were sexed and left to feed. This method was repeated for the duration of the experiment and allowed us to maintain parasitoids for several generations in the laboratory. The plants, parasitoids, and aphids were kept at 22 ± 1 °C and 65 ± 10 RH with a photoperiod of 16:8 h L:D (light:dark). Only insecticide-susceptible *M. persicae* aphids were used for rearing parasitoids.

### 2.3. Effect of Insecticides and Aphid Genotype on Patch Time Allocation in Parasitoids

LC$_{20}$ of lambda-cyhalothrin was determined for *kdr*-resistant and -susceptible *M. persicae* aphids (12.15 mg/L and 0.52 mg/L) and *A. colemani* parasitoids (0.56 mg/L) by Alfaro-Tapia et al. [39]. Hence, the sub-lethal effects of insecticides on parasitism behavior were assessed during the *A. colemani–M. persicae* interaction by comparing the patch time allocation using aphids and parasitoids exposed and unexposed to insecticide.

To treat parasitoids with dry insecticide residues, *A. colemani* females were placed for 24 h in a Petri dish (5.5 cm Ø) containing the insecticide. Water was used as control. Petri dishes were sprayed with 2 mL of insecticide or water under a pressure of 0.045 MPa using a Potter Spraying Tower (36 × 36 × 122 cm, Burkard Scientific, Middlesex, UK) and left to dry at room temperature one hour. As we expected that 20% of aphids die after insecticide application, fifteen 2nd–3rd instar nymphs of each aphid genotype were sprayed; this ensured that at least ten nymphs survived for the experiments. Aphid nymphs were sprayed with lambda-cyhalothrin according to the aphid genotype tested at the same volume and pressure as parasitoids.

The set-up for the patch allocation experiment included a leaf disk of sweet pepper (2.0 cm Ø) placed on 2% agar to prevent leaf desiccation in a plastic Petri dish (5.5 cm Ø). Exposed and unexposed parasitoid females were separately offered with resistant/susceptible and exposed/unexposed *M. persicae* aphids, following a three-way factorial arrangement (2 parasitoid conditions × 2 aphid genotypes × 3 insecticide treatments) in a completely randomized design. Our design, however, was unbalanced as susceptible aphid individuals exposed with $LC_{20}$ of the resistant aphids ($LC_{20}$ = 12.15 mg/L) were all dead, which left 10 of the 12 possible combinations.

Mummies of *M. persicae* obtained from the laboratory parasitoid colonies were isolated in Petri dishes (5.5 cm Ø) containing a drop of honey (one mummy per Petri dish). After 24 h of parasitoid emergence, individuals were sexed and allowed to mate for 24 h. Mated female parasitoids of 48 h of age (n = 25 for each condition) and without oviposition experience were individually exposed to insecticide ($LC_{20}$) or water (control) and used to study patch time allocation behavior. Petri dishes of 2.5 cm Ø containing ten wingless individual aphids from 2nd–3rd instars feeding on sweet pepper discs were offered to each parasitoid. Resistant or susceptible aphids exposed—or not exposed—to their respective $LC_{20}$ were used and represented the "patch". The parasitism behavior was recorded with a video camera HDR-CX240 (Sony, China) in these "no-choice arenas" against a white transilluminator. This set-up provides a homogeneous fluorescent white light at room temperature, for which parasitoids were previously acclimatized for two hours. We recorded 11 behavioral sequences for each treatment clustered in five groups of parameters: (1) mobility: (i) *walking outside the patch*, (ii) *walking within the patch,* and (iii) *resting*. (2) Antennal movements: (iv) *antennal contact* (contact with the aphid body of at least one antenna) and (v) *antennal examination* (contact with the aphid body between the two antennae). (3) Grooming: (vi) *antennal grooming* and (vii) *ovipositor grooming*. (4) Oviposition behavior: (viii) *sting attempt on the host* (ovipositor extruded next to an aphid in a sign of attack) and (ix) *sting*. (5) Unsuccessful test oviposition: (x) *sting attempt out of the host* and (xi) *Oviposition out of the host*. All these parameters are related to the parasitoid's infectivity [40]. The evaluation finished after 10 min of observation and the duration of each behavioral trait was analyzed with the EthoLog 2.2.5 software [41]. To discriminate between sting attempts and oviposition, we used information reported by [42]. To avoid pseudo-replications, we used new insects and new leaves for each replicate.

### 2.4. Effect of Insecticide on Parasitoid Orientation

To investigate whether insecticide interferes with the detection of chemical cues emitted by aphid-infested plants, changes in the oriented responses of female parasitoids were studied in a four-arm transparent hand-made Pettersson olfactometer [43]. Pressurized and humidified air was used to feed a central chamber through four arms at 200 mL min$^{-1}$ per arm. The borders between the four odorant branches and the decision zone were drawn on a filter paper square placed on the floor of the olfactometer. The odor field was formed in the chamber by extracting the air through the hole in the center of the floor using a Portable Phlegm Suction Unit (7E-C Yuwell). Two arms of the olfactometer were connected using silicone tubing to a glass bottle (8.25 L) containing a sweet pepper plant with ten leaves previously infested with 30 resistant or susceptible aphids. After seven days, 300–400 aphids were obtained and treated with lambda-cyhalothrin according to the aphid genotype tested or distilled water (control). The other two arms were connected to a glass bottle containing a sweet pepper plant with the same features above described but with no aphids and sprayed with distilled water (blank). To avoid any effect of the position, each arm was numbered from one to four clockwise, and the olfactometer rotated (90°) before replicating.

Mated female parasitoids (n = 25) of 72 h of age without oviposition experience and previously exposed or unexposed to the $LC_{20}$ of lambda-cyhalothrin for 24 h were placed in the central chamber of the olfactometer and left to walk freely. Before placing the plants in the odor containers, the plant's pot was carefully removed, and the roots and substrate

were placed in a round beaker glass (500 mL) and sealed airtight using Parafilm® tape and metal clamps. The position of the cue emitters was changed for each tested parasitoid, and the olfactometer was carefully washed in ethanol and dried at 150 °C before any new assay. The same experimental design described above was used for this experiment. Due to the size of the experimental system, 10 mL of insecticide (LC$_{20}$ according to the aphid genotype tested; see above) or water was hand-sprayed on sweet pepper leaves using a 1 L hand-held sprayer (Roots Garden, Easy-Chile). Changes in the position of each focal parasitoid were observed on a transilluminator for 10 min, documenting the overall time spent by each female in each arm using the event-recording software EthoLog 2.25 [38].

### 2.5. Statistical Analyses

Sub-lethal doses of insecticide on the proportion of time spent by parasitoids displaying each behavioral sequence and the ratio of the residence time of parasitoids in each olfactometer arm were analyzed using generalized mixed linear models (GLMMs) with a binomial error distribution for proportional data [44]. Insecticide exposure of parasitoids (to LC$_{20}$ or water), aphid genotype (*kdr*-resistant or susceptible), and insecticide application on aphids (low or high LC$_{20}$ or water) were used as fixed effects. The number of replicates and the number of assays were included as random effects [45]. Models with and without interactions between fixed effects were compared using the Akaike criterion. In all cases, the model with interactions was chosen. This analysis was conducted to account for complex data structures with various levels and deal with unbalanced datasets [46,47]. Unbalanced data were produced due to all susceptible aphids dying when sprayed with the LC$_{20}$ for the *kdr*-resistant genotype. We used a Tukey test corrected for multiple comparisons by the single-step method using the Multcomp package [48] to establish significant differences. All the analyses were conducted using the *lme4* package [49] in R version 3.5.1 [50].

## 3. Results

### 3.1. Effect of Insecticides and Aphid Genotype on the Time Allocation of Parasitoids

First, the mobility of *A. colemani* parasitoids was differentially affected by their exposure to insecticide, by the genotype of aphids, and by the application of insecticide on aphids. The proportion of time spent walking out of the patch ($\chi^2 = 1.92$; $df = 1$; $p = 0.17$), walking in the patch ($\chi^2 = 0.01$; $df = 1$; $p = 0.93$), and resting ($\chi^2 = 0.08$; $df = 1$; $p = 0.78$) was not significantly affected by the exposure of parasitoids to LC$_{20}$ of lambda-cyhalothrin. In contrast, an effect of the genotype was observed in the time spent walking out of the patch ($\chi^2 = 7.49$; $df = 1$; $p = 0.01$), mainly affected by the insecticide concentration ($\chi^2 = 1435.96$; $df = 2$; $p < 0.001$) compared with parasitoids faced with susceptible and *kdr*-resistant aphids exposed and unexposed to low LC$_{20}$ = 0.52 mg/L of susceptible *M. persicae* (Figure 1A). Although the time spent walking in the patch was not affected by the aphid genotype ($\chi^2 = 2.73$; $df = 1$; $p = 0.10$), the time spent by parasitoids walking in the patch decreased in sprayed *kdr*-resistant aphids with high LC$_{20}$ ($\chi^2 = 2063.23$; $df = 2$; $p < 0.001$) (Figure 1B). Similarly, the resting time was not significantly affected by the aphid genotype ($\chi^2 = 1.88$; $df = 1$; $p = 0.17$), but it increased in sprayed resistant aphids ($\chi^2 = 16.52$; $df = 2$; $p < 0.001$) (Figure 1C) without interactions between factors.

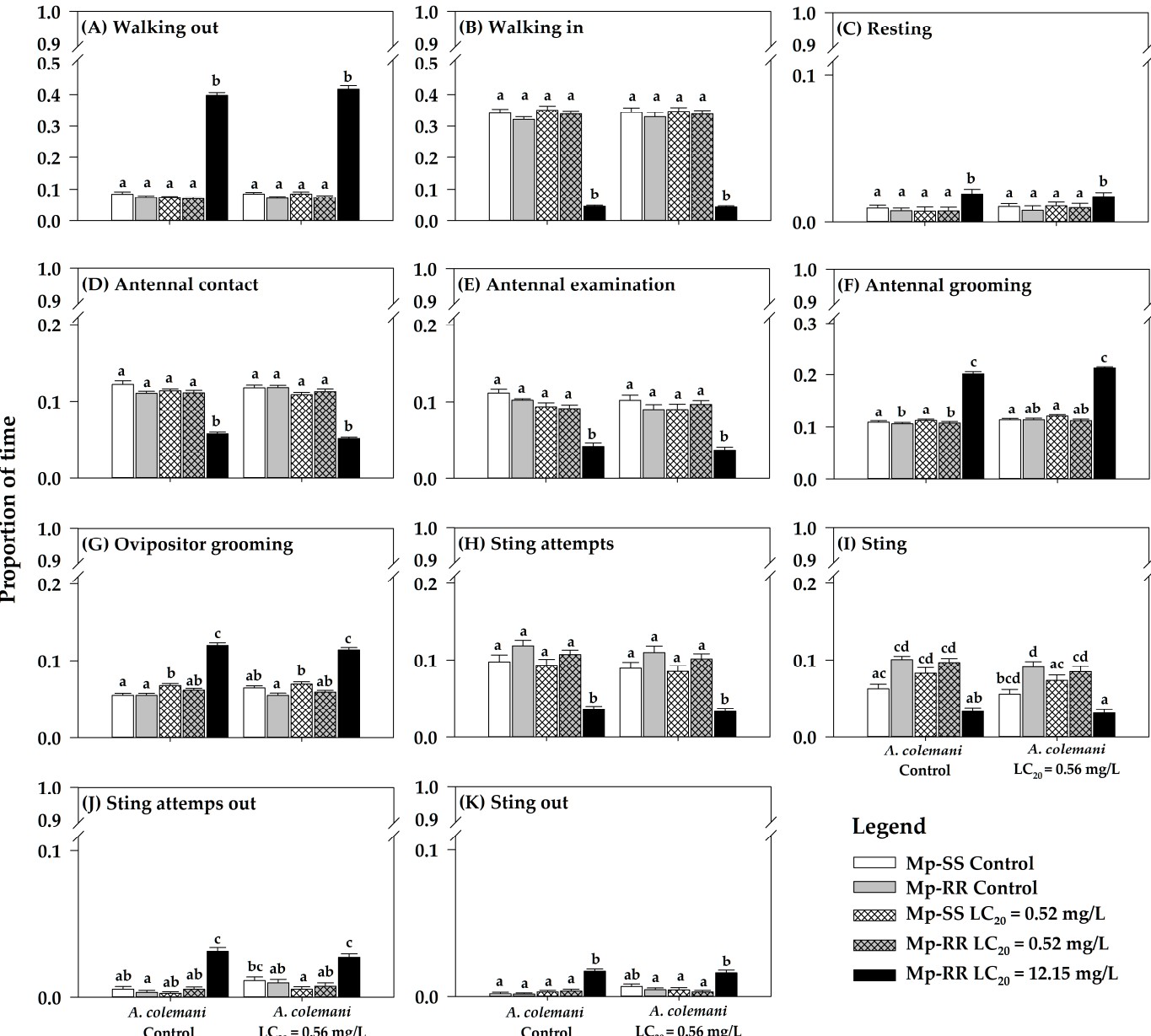

**Figure 1.** Proportion of time spent showing different behaviors in individuals of the parasitoid wasp *Aphidius colemani* exposed and unexposed to the insecticide lambda-cyhalothrin (Zero 5 EC). Parasitoid individuals were offered with *kdr*-susceptible (Mp-SS) and *kdr*-resistant (Mp-RR:) *M. persicae* aphids exposed (LC$_{20}$) and unexposed (water as control) to lambda-cyhalothrin. Mp-SS exposed to LC$_{20}$ of *kdr*-resistant aphids are not shown, as all aphids were dead. Different letters indicate significant differences (Tukey multiple comparisons test $p < 0.05$).

Second, the group of behavioral parameters related to antennal movements were affected not by the exposure of parasitoids to insecticide or the aphid genotype, but the insecticide application on aphids. The exposure of parasitoids to LC$_{20}$ did not change the proportions of time spent in antennal contact ($\chi^2 = 0.55$; $df = 1$; $p = 0.46$) and antennal examination ($\chi^2 = 2.87$; $df = 1$; $p = 0.09$), regardless of the aphid genotype to which parasitoids were confronted ($\chi^2 = 1.02$; $df = 1$; $p = 0.31$ and $\chi^2 = 0.15$; $df = 1$; $p = 0.69$, respectively). However, the time spent in antennal contact ($\chi^2 = 587$; $df = 2$; $p < 0.001$) and antennal examination ($\chi^2 = 146.18$; $df = 2$; $p < 0.001$) decreased when parasitoids were faced to *kdr*-resistant aphids exposed to high LC$_{20}$ of resistant *M. persicae*. Marginal significant interactions between exposed parasitoid and aphid genotype ($\chi^2 = 3.95$; df = 1; $p = 0.05$) and the interaction between exposed parasitoid and sprayed aphid ($\chi^2 = 6.32$; $df = 2$; $p = 0.04$)

were observed (Figure 1D) regarding the proportion of time spent in antennal contacts. However, there was no significant interaction detected between exposed parasitoid, aphid genotype, and sprayed aphid factors for antennal examination (Figure 1E).

Third, the parasitoid wasp *A. colemani* spent more time in grooming behaviors depending on the aphid genotype and aphid exposure to insecticide. The time used in antennal grooming (Figure 1F) increased ($\chi^2$ = 17.74; *df* = 1; *p* < 0.001) in parasitoids exposed to $LC_{20}$, but no changes were observed in the time spent in ovipositor grooming ($\chi^2$ = 0.33; *df* = 1; *p* = 0.56) (Figure 1G). However, when the parasitoids were faced with *kdr*-resistant aphids, both the antennal grooming and the ovipositor grooming lasted longer, regardless of whether the aphids were sprayed or not ($\chi^2$ = 5.48; *df* = 1; *p* = 0.02 and $\chi^2$ = 15.13; *df* = 1; *p* < 0.001, respectively). In addition, both behaviors were affected by the exposure to high $LC_{20}$ of *kdr*-resistant aphids ($\chi^2$ = 1454.54; *df* = 2; *p* < 0.001 and $\chi^2$ = 600.08; *df* = 2; *p* < 0.001, respectively), without interactions between factors. Fourth, the time spent on oviposition behaviors significantly decreased in parasitoids faced with resistant aphid genotypes. Although the time used in sting attempts ($\chi^2$ = 1.27; *df* = 1; *p* = 0.26) and stings into aphids ($\chi^2$ = 2.65; *df* = 1; *p* = 0.10) were not affected in parasitoids exposed to $LC_{20}$, the proportion of time spent on sting attempts ($\chi^2$ = 5.94; *df* = 1; *p* = 0.02) and stings ($\chi^2$ = 11.18; *df* = 1; *p* < 0.001) significantly decreased when parasitoids were confronted with *kdr*-resistant aphids (Figure 1H,I). Similarly, this trend was observed when the parasitoids were faced with sprayed resistant aphids exposed to high $LC_{20}$ of resistant *M. persicae* (sting attempts: $\chi^2$ = 78.31; *df* = 2; *p* < 0.001 and stings: $\chi^2$ = 66.31; *df* = 2; *p* < 0.001) as compared to non-sprayed aphids and sprayed with low $LC_{20}$ of susceptible *M. persicae*. No significant interactions between factors were observed.

Finally, the time used in unsuccessful test oviposition increased in the parasitoids exposed to $LC_{20}$ doses. This was observed by an increase in the proportion of time spent in sting attempts out of the host ($\chi^2$ = 5.88; *df* = 1; *p* = 0.02) and stings out of the host ($\chi^2$ = 4.12; *df* = 1; *p* = 0.04). Similarly, the time spent in sting attempts out of the host ($\chi^2$ = 56.79; *df* = 2; *p* < 0.001) and stings out of the host ($\chi^2$ = 47.50; *df* = 2; *p* < 0.001) was significantly affected in parasitoids faced with sprayed resistant aphids exposed to high $LC_{20}$ (Figure 1J,K). Furthermore, a significant interaction between the exposure of parasitoids to insecticides and insecticide application on aphids was observed for the proportion of time spent on stings out of the host ($\chi^2$ = 8.68; *df* = 2; *p* = 0.01). All GLMM best-mixed models with binomial error and logit link function developed for these variables are shown in Table S1.

### 3.2. Influence of Insecticide Exposure on the Orientation Behavior of Parasitoids

The effects of insecticides on orientation responses in *A. colemani* were observed when exposed and unexposed parasitoids were faced with susceptible and *kdr*-resistant aphids applied and not treated with insecticide. The results show that the proportion of time parasitoids spent on arms that contained sweet pepper leaves with aphids did not change with the exposure to the $LC_{20}$ of lambda-cyhalothrin ($\chi^2$ = 3.03; *df* = 1; *p* = 0.08) or the aphid genotype ($\chi^2$ = 0.91; *df* = 1; *p* = 0.33). This means that the attraction for cues emitted from aphid-infested sweet pepper leaves is not affected by the exposure of parasitoids to insecticide. The attraction cues seem not to change with the aphid genotype. However, parasitoids challenged with aphids treated with insecticide spent a significantly lower proportion of time in arms containing sweet pepper leaves with aphids ($\chi^2$ = 894.00; *df* = 2; *p* < 0.001). This latter result suggests that parasitoid females are less attracted (or repelled) by *M. persicae* exposed to the $LC_{20}$ for the resistant aphid genotype concerning the remaining combination of treatments (Figure 2).

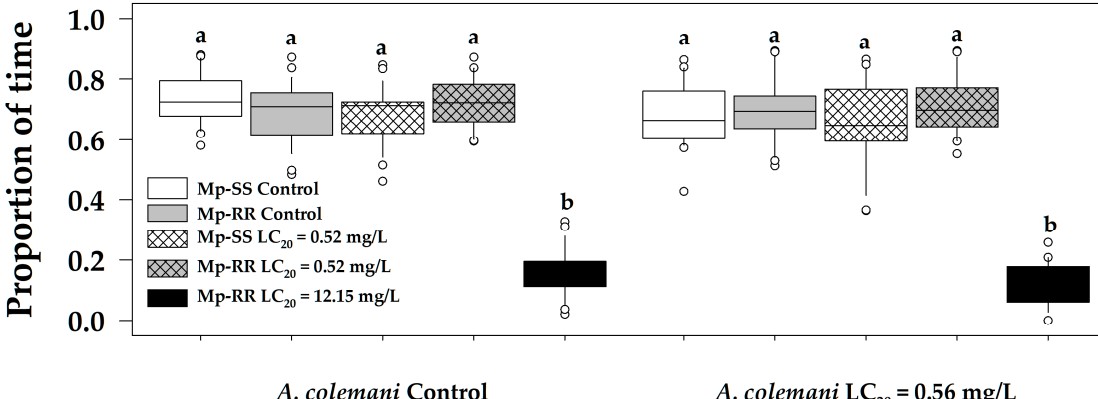

**Figure 2.** Proportion of time spent on each arm of the olfactometer in individuals of the parasitoid wasp *Aphidius colemani* exposed and unexposed to $LC_{20}$ of the insecticide lambda-cyhalothrin (Zero 5 EC). Using a 4-arm olfactometer, parasitoid individuals were offered with *kdr*-susceptible (Mp-SS) and *kdr*-resistant (Mp-RR:) *M. persicae* aphids exposed ($LC_{20}$) and unexposed (water as control) to lambda-cyhalothrin. Mp-SS exposed to $LC_{20}$ of kdr-resistant aphids are not shown, as all aphids were dead. Two arms of the olfactometer were connected to glass bottles containing pepper plants with aphids, while the other two arms were connected to glass bottles containing only pepper plants (no aphids). Different letters indicate significant differences (Tukey multiple comparisons test, $p < 0.05$).

## 4. Discussion

Many studies have separately addressed the sub-lethal effects of insecticides on parasitoid behavior [15,17,21,51,52] and resistant aphids [9,53–55]. In this work, we assessed how their exposure could alter the interaction between the aphid parasitoid *A. colemani* and its aphid host *M. persicae* to residual insecticides under laboratory conditions. Interestingly, our results show that the studied parasitism behavior, in terms of the proportion of time parasitoids invest in orientating to and manipulating their hosts, seems not to be affected by previous exposure to residual insecticides. Instead, parasitoids seem reluctant to interact with *kdr*-resistant aphids sprayed with high concentrations of insecticides, an effect that disappears when aphids are sprayed with low doses.

### 4.1. Parasitoids Avoid Parasitizing Insecticide-Resistant Aphids at High Doses

The decreased mobility, exploration, oviposition, and grooming behaviors observed in parasitoids exposed and unexposed to their sub-lethal concentrations, and faced with *kdr*-resistant aphids treated with high doses of insecticides suggest that aphid genotype may play a major role in defining the outcome of parasitism in agroecosystems, where insect pests are mainly managed with insecticides. Indeed, the stronger the intensity of insecticide selection, the higher the frequency of insecticide-resistant individuals and the lower the efficacy of biological control, which can be expected if insecticides are frequently applied at higher doses, lethal for aphid parasitoids. Two hypotheses can explain our results: (1) pyrethroids, such as lambda-cyhalothrin, are neurotoxic insecticides [56]. Therefore, they can potentially perturb behaviors such as host searching, which leads the female parasitoids to avoid high concentrations of insecticides and, hence, not parasitize the *kdr*-resistant aphids. (2) Some of the coadjuvant ingredients used to prepare lambda-cyhalothrin can rapidly volatilize at room temperature [57]. Thus, pyrethroids seem to have a repellent effect against *A. colemani* when parasitoid wasps are offered with *kdr*-resistant aphids sprayed with high concentrations of lambda-cyhalothrin. It is possible that the effective detection of the aphid host by *A. colemani* was disturbed by high concentrations of lambda-cyhalothrin since oviposition involves sensory signals based on physical and chemical cues (i.e., olfactory and gustatory) acting at a short-range [58,59]. It could be that a masking odor interferes with host localization or decreases the attractiveness of the host. It has been reported that insecticides may negatively affect natural enemies over a short time scale, from the insecticide application to residual exposure during patch residence time, exploration, and oviposition [21,60,61]. Similarly, increased grooming behavior may

be a cause or consequence of increased insecticide exposure when parasitoids are faced with *kdr*-resistant aphids treated with high concentrations of insecticides, probably because high concentrations of insecticide stimulate the chemo- and mechanoreceptors [62]. A significant event to notice is that in our experiment, *A. colemani* showed in some cases detrimental oviposition behavior, observed in events of sting-attempts and stings on leaf surfaces and outside the host; however, this disturbance is not reflected in the rate of parasitism, productivity, and survival of the offspring [39]. On the other hand, alterations in the cuticle structure may result in slower insecticide absorption in resistant aphids [5]; however, we did not find evidence that the parasitoids' behavior changes related to that. Therefore, the observed contact avoidance behavior changes in parasitoids seem more likely associated with high LC$_{20}$ than carrying *kdr* mutation in resistant *M. persicae* aphids.

These results are confirmed by the response of *A. colemani* in our orientation bioassays. When parasitoids were faced with high doses of insecticide, they showed a reduction in the time spent on the source of infested plants with the *kdr*-resistant aphids treated with their sub-lethal concentrations. An explanation for this effect is that pyrethroid insecticides act like a repellent similar to those observed in mosquitoes [63] and honeybees [64]. However, this repellency has been found in contact bioassays with pyrethroid residues on treated surfaces rather than with pyrethroids in a volatile fraction that the parasitoids could perceive through the airflow in an olfactometer test. The vapor pressure of lambda-cyhalothrin is low compared with other insecticides, and consequently, it is unlikely that it could act as a volatile repellent at laboratory temperatures [57]. Another explanation is that adjuvants or additives in our experiment's formulated insecticide could be responsible for the repellent effect observed at higher concentrations [65]. Finally, the repellency found on parasitoids could also be explained as reducing the emission of attractive volatiles by the aphid feeding activity on the plant or the plant itself [66]. The sub-lethal insecticide concentration used on resistant aphids did not produce mortality. Still, it could reduce feeding activity that could lower the production of volatile organic compounds by the injured plants. In any case, repellency is associated with sensory perception [64,67], allowing the parasitoid to perceive high insecticide concentrations after contact and avoid treated areas since pyrethroids have been recorded to exhibit higher acute lethal toxicity to *A. colemani* [25]. Therefore, the vulnerability of *kdr*-resistant *M. persicae* could not be capitalized by this parasitoid species under a high concentration of lambda-cyhalothrin, causing a significant reduction in parasitism rates, survival, and longevity [39]. Additionally, pyrethroids target ion channels involved in central nervous system function, where their primary mode of action is to interfere with the normal functioning of voltage-gated sodium channels [68]. Consequently, this odor detection process and the subsequent parasitoid behavior responses depend on neural transmissions [16], which are expected to be affected by neurotoxic insecticides, including lambda-cyhalothrin.

*4.2. Non-Detrimental Effects of Low Insecticide Concentrations*

Contrary to what we expected, this study demonstrated no effect of sub-lethal concentrations applied to *A. colemani* faced with susceptible and *kdr*-resistant *M. persicae* exposed to sub-lethal concentrations (low concentrations). This result may be because sub-lethal concentrations of both parasitoids and susceptible aphids were similar. After exposure to lambda-cyhalothrin, the parasitoids tested did not exhibit alterations in the proportion of time spent when evaluated different behaviors and female parasitoids continue to parasitize aphids.

These results are similar to what was found in other species of Aphidiinae (Hymenoptera), where it has been shown that pyrethroids did not modify parasitoid behavior [15,16]. In our study, an increase in mobility, exploration, oviposition, and decreased grooming behavior were observed. Similar results have been found for *Aphidius ervi*, *A. matricariae,* and *Diaeretiella rapae* (Hymenoptera, Braconidae) [15,16,24] after 24 h of exposure, suggesting that in this period, parasitoids can compensate for the sub-lethal effects of pyrethroids. The increase in oviposition behavior may be due to a decrease in

treated susceptible *M. persicae* mobility or complete aphid immobility, given that parasitism on treated susceptible aphids should be easier for the parasitoid than susceptible aphids not treated with insecticide, which showed higher mobility. These results are corroborated by Cabral et al. [69], where a decrease in the aphid mobility resulting from insecticide treatment increased the voracity of *Coccinella undecimpunctata* (Coleoptera, Coccinellidae).

Additionally, the longer amount of time spent in sting attempts and stings on susceptible *M. persicae* exposed to insecticides and on *kdr*-resistant *M. persicae* exposed and unexposed to low concentrations of insecticide, may be due to the low response to the alarm pheromone of these aphid species in comparison to susceptible *M. persicae* [39]. These results are supported by the evidence that *kdr*-insecticide resistance mechanisms have inhibitory, pleiotropic effects on aphid responses to alarm pheromone (E)-ß-farnesene [7,54] and that reduced alarm response is directly attributable to the mechanism of insecticide resistance [9] and vulnerability to parasitoids [7]. Moreover, aphid mobility and defensive behaviors can influence prey selection for the parasitoid, given that aphids are known to exhibit a wide range of defensive behaviors such as kicking, attacking with their legs, and walking away [70]. Additionally, this study showed that *A. colemani* of both conditions were attracted by the odor of this host–plant complex treated and untreated with low concentrations of insecticides, demonstrating that orientation towards aphid-infested plants was due mainly to the release of odors induced by aphids than insecticide application. Similar results were observed in *Aphidius ervi* (Hymenoptera, Braconidae), where the orientation was not perturbed by the application of lambda-cyhalothrin at sub-lethal doses [15].

However, contrary to our results, some studies reported that pyrethroids disrupt locomotor function and parasitoids' physical perception [20,71,72]. For instance, pyrethroids insecticides at sub-lethal doses could disturb some behavior parameters, decreasing the attack rates of *Trissolcus semistriatus* (Hymenoptera, Platygastridae) [73], the walking speed in *Trissolcus basalis* (Hymenoptera, Platygastridae) [20], and the oviposition behavior of *A. ervi* parasitizing *M. persicae* on oilseed rape [52]. An explanation of our results could be that parasitoid wasp rearing began from the mummies of *M. persicae* collected on peach orchards. Pyrethroids and neonicotinoids are regularly applied to these fields. Hence, more resistant hosts could accelerate parasitoid resistance due to the exposure of the parasitoid larvae to insecticide selection through their hosts [74]. Lower target site sensitivity and enhanced detoxification may be potential causes of lower toxicity of lambda-cyhalothrin to *A. colemani* [25].

### 4.3. Implications for Integrated Pest Management (IPM)

It is known that the infectivity of parasitoids depends on host selection behavior, physiological adaptations, and the parasitoids' skills to determine the availability of their aphid host and the risks involved in obtaining that resource [75]. The low performance of *A. colemani* faced with *kdr*-resistant *M. persicae* treated with high insecticide concentrations and the opposite response against susceptible and *kdr*-resistant *M. persicae* treated with low doses offer a window for its implementation in IPM programs. Indeed, the integration of chemical and biological methods needs to consider how insecticides impact natural enemies of target pests. Under field conditions, parasitoids are exposed to high concentrations of different pesticides, decreasing as the insecticide degrades [76], which might not cause lethal effects on parasitoids. For instance, when *A. colemani* is exposed to cypermethrin, parasitoids are negatively affected almost immediately on the same day of application, the effects being reduced after three days of application [77]. Our results suggest that *A. colemani* could be introduced a few days after application to obtain a high survival rate. They can withstand lambda-cyhalothrin residues since, under field conditions, it is well known that pyrethroids have a short biodegradation period [78,79].

Nevertheless, it is important to note that the present study was conducted in a laboratory where parasitoids and aphids were subjected to controlled conditions. Toxicity data extrapolation of laboratory bioassays in parasitoids to field conditions could be problematic and probably lead to overestimating actual hazards [80]. However, the success of

parasitoids as biological control agents in agroecosystems may fail when factors that decrease the parasitism performance of these natural enemies are present in the environment. Therefore, broad-spectrum insecticides are one of the challenging constraints associated with IPM implementation [21].

## 5. Conclusions

Usually, insecticide application and the biological control of pests are incompatible, but this is not always the case. Our results demonstrated that short-term pesticide exposure at high concentrations causes adverse effects, indicating that insecticides may block some physiological processes, leading to the disruption of the behavior and orientation of *A. colemani*. Our findings demonstrate the sub-lethal effects of insecticides on the aphid parasitoids and contribute to a better understanding of how residual insecticides may impact the biological performance of natural enemies, providing an insight into the interaction between pests, natural enemies, and insecticides. However, studies in actual field conditions are needed to confirm our results.

**Supplementary Materials:** The following are available online at https://www.mdpi.com/article/10.3390/agriculture11060539/s1, Table S1: Generalized linear mixed models (GLMM) showing the full model evaluated for each behavior observed in *Aphidius colemani* (Ac) individuals exposed and unexposed to $LC_{20}$ of lambda-cyhalothrin (Ac condition: AcC) and faced with susceptible and *kdr*-resistant *M. persicae* genotypes (MpG) exposed and unexposed to their respective $LC_{20}$ of lambda-cyhalothrin (Zero 5 EC) (Mp condition: MpC). For each level, the Chi-square statistical test ($\chi^2$), the degrees of freedom (*df*), and the *p*-value are represented.

**Author Contributions:** Conceptualization, A.A.-T., E.F.-C., and C.C.F.; methodology, A.A.-T., and J.K.A.-B.; software, A.A.-T.; validation, A.A.-T., J.K.A.-B., E.F.-C., and C.C.F.; formal analysis, A.A.-T., E.F.-C., and C.C.F.; resources, C.C.F.; data curation, A.A.-T., J.K.A.-B.; writing—original draft preparation, A.A.-T., E.F.-C., and C.C.F.; writing—review and editing, A.A.-T., J.K.A.-B., E.F.-C., and C.C.F.; supervision, E.F.-C., and C.C.F.; project administration, C.C.F.; funding acquisition, C.C.F. All authors have read and agreed to the published version of the manuscript.

**Funding:** This work was funded by Iniciativa Científica Milenio grant NC120027 from the Chilean Ministry of Economy. A.A.-T and J.K.A.-B. acknowledge fellowships for M.Sc. studies granted by the Millennium Nucleus Centre in Molecular Ecology and Evolutionary Applications in Agroecosystems. A.A.-T. and J.K.A.-B. received support from FONDECYT grant 1130483 to C.C.F. to complete their Master's theses. Additionally, the authors received funding from FONDECYT regular grant 1170943 to C.C.F. during the preparation and writing of this manuscript.

**Institutional Review Board Statement:** Not applicable.

**Informed Consent Statement:** Not applicable.

**Data Availability Statement:** The data is available upon request from the corresponding author.

**Acknowledgments:** The authors would like to thank Lucia Briones for her help in molecular analysis. We are also grateful to Cristian Baltierra Valenzuela (Fundo El Amanecer) for his valuable support in the sampling of parasitoids.

**Conflicts of Interest:** The authors declare no conflict of interest.

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
