# Peer review of "Biological Control May Fail on Pests Applied with High Doses of Insecticides: Effects of Sub-Lethal Concentrations of a Pyrethroid on the Host-Searching Behavior of the Aphid Parasitoid Aphidius colemani (Hymenoptera, Braconidae) on Aphid Pests"

_agriculture, doi:10.3390/agriculture11060539_

Round 1

Reviewer 1 Report

This manuscript describes results obtained from a detailed laboratory study of the impact of sub-lethal concentrations of the pyrethroid lambda-cyhalothrin on the behaviour of the parasitoid Aphidius colemani. The results are, by nature of the complex interaction of different behavioural responses and susceptible versus resistant aphids, quite difficult to summarise and interpret. However, this is a well written and well considered manuscript that I recommend  minor edits.

The authors rightly highlight the need to verify findings in field trials but the results have important implications that would assist with the design of such experiments as well as contributing to the field of IPM related research.

Minor recommended revisions

General

  1. Introduction: references 1 and 2 are very old (2007) -can the authors include more recent publications as these are key to the premise of the paper.

  1. Introduction and discussion

Can the authors include information about kdr resistance mechanisms? – whilst predominantly mutations in ion channels there are also reports of differences in cuticular proteins and composition (eg. Silva et al 2012 PLOs one). I do wonder if this may have an impact upon parasitoid behaviour in this study and recommend the authors include consideration of this aspect in the discussion section.

  1. 2.2 “Parasitoids were obtained by sampling M. persicae on peach trees” -later in the manuscript the authors highlight that these trees are treated with insecticides (including pyrethroids) -this should be highlighted in section 2.2 (as it is a key point) and if possible the orchard treatment regime (eg. how many times p.a. are the trees sprayed and if this is considered to be a “normal/typical” treatment regime)

  1. line 290 parasitoid should be singular not plural
  2. line 359 delete “the” before central nervous system
  3. line 372 delete “evaluated” or place before behaviours

Author Response

Response to reviewer 1 (comments are reproduced for clarity):

This manuscript describes results obtained from a detailed laboratory study of the impact of sub-lethal concentrations of the pyrethroid lambda-cyhalothrin on the behaviour of the parasitoid Aphidius colemani. The results are, by nature of the complex interaction of different behavioural responses and susceptible versus resistant aphids, quite difficult to summarize and interpret. However, this is a well written and well considered manuscript that I recommend minor edits.

The authors rightly highlight the need to verify findings in field trials but the results have important implications that would assist with the design of such experiments as well as contributing to the field of IPM related research.

We want to thank the reviewer for the positive and insightful comments on the manuscript. Below is our response to the subjects raised in the review.

The Introduction: references 1 and 2 are very old (2007) -can the authors include more recent publications as these are key to the premise of the paper.

References 1 and 2 (lines 39 and 40) were updated accordingly (lines 544-568).

Can the authors include information about kdr resistance mechanisms? – whilst predominantly mutations in ion channels there are also reports of differences in cuticular proteins and composition (eg. Silva et al 2012 PLOs one). I do wonder if this may have an impact upon parasitoid behaviour in this study and recommend the authors include consideration of this aspect in the discussion section.

  • The main mechanisms involved in insecticides resistance in aphids are (1) detoxification, through alteration in the enzyme activities that degrade or sequester insecticides, (2) insensitivity, due to point mutations in genes encoding for proteins that are the target site of insecticides, and (3) and alterations in the outer cuticle structure (Silva et al 2012). For a better understanding, we added relevant information and paraphrased this part (41-53).
  • However, as we indicated in the Discussion, we did not find evidence that aphids' slow insecticide absorption can affect the parasitoid behavior. We added a paragraph in the Discussion (line 417-422). The reference above mentioned is listed in lines 576-577

2.2 "Parasitoids were obtained by sampling M. persicae on peach trees" -later in the manuscript the authors highlight that these trees are treated with insecticides (including pyrethroids) -this should be highlighted in section 2.2 (as it is a key point) and if possible, the orchard treatment regime (eg. how many times p.a. are the trees sprayed and if this is considered to be a "normal/typical" treatment regime)

Usually, Chilean farmers make regular applications on peach orchards under conventional management. Constant monitoring is done since flowering, mainly at the beginning of spring when aphids colonize peach trees (Abarca et al 2017). It is recommended to make applications only with those insecticides registered by the Servicio Agrícola "SAG" that include pyrethroids and neonicotinoids. The same relevant information was added in the sub-section "2.2 Parasitoids" (lines 132-136) as suggested, and the reference listed.

line 290 parasitoid should be singular not plural

Corrected (line 366)

line 359 delete "the" before central nervous system

Corrected (line 446)

line 372 delete "evaluated" or place before behaviours

Corrected (line 458)

Reviewer 2 Report

Although the goal of the study is sound, there are important weaknesses in Methods, Results and Discussion. The experimental design is not well described. How does the olfactometer look like? Self-made or commercial? Does the citation include a technical chart? Four arms numbered clockwise beginning from North? Can you test whether the magnetic orientation is present? Exposure to insecticide is not described. Baker glass - perhaps beaker?

The first paragraph of Results looks like if there were four combinations, but in the graph, there are five! The Method part of manuscript does not describe the really used method well, it does not mention 5 combinations. The graph probably shows that a high dose of cyhalothrin functions as a repellent, while aphid genotype had a minor effect. This Result text and Table S1 does not correspond to the graphs. Thus, the entire manuscript must be re-written. 

Discussion: Sentence "effect that appears to be enhanced when aphids are sprayed with low doses." is false - low doses did not make any effect.

The title "Parasitoids avoid parasitizing insecticide-resistant aphids" is wrong. Parasitoids in fact avoided ovipositing in aphids sprayed by high doses.

The suggestion that "effective detection of the aphid host by A. colemani was disturbed by high concentrations" is possible but I do not think it is a blocked detection. I think it is a repellent effect of the chemical.

Caption for Table S1 is not consistent with the table abbreviations. 

Author Response

Reviewer 2:

Although the goal of the study is sound, there are important weaknesses in Methods, Results and Discussion. The experimental design is not well described. How does the olfactometer look like? Self-made or commercial? Does the citation include a technical chart? Four arms numbered clockwise beginning from North? Can you test whether the magnetic orientation is present? Exposure to insecticide is not described. Baker glass - perhaps beaker?

Many thanks for your valuable comments and suggestions to improve the manuscript. We have restructured the manuscript and included your suggestions.

  • For a better understanding, the experimental design was included in the subsection "2.3 Effect of insecticides and aphid genotype on patch time allocation in parasitoids" (lines 181-189).
  • The olfactometer was hand-made following the specifications from Vet et al (1983). The set-up for the olfactometry assays is shown in the following figure. A four-pointed exposure chamber was constructed of four perspex crescents (90°arc and radius 135mm), glued to the 3mm thick perspex sheet forming the ceiling. At each point of the star, a 5mm diameter hole is formed. This array was then firmly held down on the perspex floor sheet using four metal clamps. Air leaks were prevented using Teflon tape of 0.1 mm thick (topex-Chile). The odor field was formed in the chamber by extracting the air through the hole in the center of the floor. The outlets of each glass jar were connected with silicone tubing to inlets of the four-way olfactometer. Relevant information was added in sub-section 2.4 Effect of insecticide on parasitoid orientation (lines 218-237).

A four-way olfactometer was used for orientation tests. A constant airflow allows the diffusion of olfactory stimuli in each arm connected to jars with infested plants and control. The square-shaped "decision zone" in the center of the olfactometer is considered a zone of indeterminate choice. After letting odors diffuse, parasitoids were placed in the center of the olfactometer through a hole and observed for 600 s.

  • The cite from Vet and Godfray (2008) was changed by Vet et al (1983) (line 220) and included in the list of reference (lines 658-662)
  • The four arms of the olfactometer were numbered clockwise, beginning from North in each replicate.
  • We did not test any influence of the magnetic orientation. However, to avoid any effect of the position, each arm was numbered from one to four clockwise, and the olfactometer rotated (90°) before replicating. This part is detailed in lines 235-237.
  • For a better understanding, the exposure to insecticide was included in the sub-section: 3 Effect of insecticides and aphid genotype on patch time allocation in parasitoids (lines 165-174), and it was mentioned in the sub-section 2.4 Effect of insecticide on parasitoid orientation (lines 251-256).
  • The wordbaker” was changed by “beaker” (line 248)

The first paragraph of Results looks like if there were four combinations, but in the graph, there are five! The Method part of manuscript does not describe the really used method well, it does not mention 5 combinations. The graph probably shows that a high dose of cyhalothrin functions as a repellent, while aphid genotype had a minor effect. This Result text and Table S1 does not correspond to the graphs. Thus, the entire manuscript must be re-written.

  • As we mentioned above, the experimental design was included in the subsection "2.3 Effect of insecticides and aphid genotype on patch time allocation in parasitoids" (lines 181-189). We now explain the combinations and the factorial design used.
  • We added a paragraph in the discussion to discuss the effect of repellency (lines 417-412) in addition to the information already detailed in the same section (lines 426-435).
  • The results described in the sub-section "1. Effect of insecticides and aphid genotype on the time allocation of parasitoids" were corroborated according to table S1. In the same way, we rephrased the entire section for a better understanding of our findings. We found an error in the P-value corresponding to an interaction described for the variable "grooming ovipositor" between the condition of the parasitoid and the genotype of the aphid. We corrected the referenced wrong value of 0.03 for the correct value of 0.08, so this paragraph was removed and corrected (lines 320-326).

Discussion: Sentence "effect that appears to be enhanced when aphids are sprayed with low doses." is false - low doses did not make any effect.

Thanks for your suggestion. This part was rephrased for a better understanding. (lines 384-385)

The title "Parasitoids avoid parasitizing insecticide-resistant aphids" is wrong. Parasitoids in fact avoided ovipositing in aphids sprayed by high doses.

Changed according to your suggestion: Biological control may fail on pests applied with high doses of insecticides: Effects of sub-lethal concentrations of a pyrethroid on the host-searching behavior of the aphid parasitoid Aphidius colemani (Hymenoptera, Braconidae) on aphid pests.

The suggestion that "effective detection of the aphid host by A. colemani was disturbed by high concentrations" is possible but I do not think it is a blocked detection. I think it is a repellent effect of the chemical.

This paragraph is paraphrased in the discussion explaining the repellent effect (lines 399-40; 417-422 and lines 426-435).

Caption for Table S1 is not consistent with the table abbreviations.

It was done according to your suggestion (lines 540-545).

Round 2

Reviewer 2 Report

The manuscript is now much improved. However, one methodological issue remained unresolved. This not compromising the results and conclusions. It just must be corrected for clarity and repeatability. It concerns the description of the design of treatments. On line 186, the authors say: the arrangement was 2x2x3 conditions. But there were not all 12 treatments, only 10. (Which makes sense, in the missing treatments all aphids would be dead.) So, change the description on line 186 and some connected sentences over an entire manuscript. Mainly the sentences mentioning the concentration of insecticide "respective" to the aphid strain. It was not so. Respective would be low concentration for susceptible aphids and high for resistant aphids. But there were also resistant aphids with low concentration. This is good, the results in tables and graphs are correct. Only some sentences are not. There are inconsistent sentences in chapter 3.1. And strangely, also strange statistical tests. There are tests with df=1, but better should be tests that compared all five combinations shown in graphs, it is df=4. There also are highlighted some more minor corrections needed in the attached file. I also marked some sections that I consider very good. 
